# Clinical, Laboratory, and Imaging Findings of Pregnant Women with Possible Vertical Transmission of SARS-CoV-2—Case Series

**DOI:** 10.3390/ijerph191710916

**Published:** 2022-09-01

**Authors:** Marius Craina, Daniela Iacob, Mirabela Dima, Sandor Bernad, Carmen Silaghi, Andreea Moza, Manuela Pantea, Adrian Gluhovschi, Elena Bernad

**Affiliations:** 1Department of Obstetrics and Gynecology, “Victor Babes” University of Medicine and Pharmacy, Eftimie Murgu Square No. 2, 300041 Timisoara, Romania; 2Clinic of Obstetrics and Gynecology, County Clinical Emergency Hospital “Pius Brinzeu”, 300723 Timisoara, Romania; 3Department of Neonatology, “Victor Babes” University of Medicine and Pharmacy, Eftimie Murgu Square No. 2, 300041 Timisoara, Romania; 4Romanian Academy Timisoara Branch, Mihai Viteazul Avenue, 24, 300275 Timisoara, Romania

**Keywords:** pregnancy, clinical, laboratory, imaging, newborn, SARS-CoV-2, COVID-19, vertical transmission

## Abstract

The Severe Acute Respiratory Syndrome Coronavirus 2 (SARS-CoV-2) pandemic significantly impacted the general population’s health. At times, the infection has unfavorably influenced pregnancy evolution and the result of birth. However, vertical transmission of the virus is rare and generates controversial discussions. The study aimed to highlight the clinical, laboratory, and imaging findings of pregnant women with confirmed Coronavirus Disease 2019 (COVID-19) with possible vertical transmission and identify possible factors that encourage vertical transmission. Between 1 April 2020 and 31 December 2021, 281 pregnant women diagnosed with COVID-19 gave birth in the Obstetrics and Gynecology Departments of the tertiary unit of County Emergency Clinical Hospital from Timisoara. Three newborns (1.06%) tested positive. The characteristic of these three cases was described as a short series. In two cases, the patients were asymptomatic. In one case, the patient developed a mild form of COVID-19 with a favorable evolution in all cases. We did not identify the presence of smoking history, vaccine before admission, atypical presentation, fever, or chest X-ray abnormalities. We note possible factors that encourage vertical transmission: Pregnancy-induced hypertension, thrombophilia, asymptomatic cough, an asymptomatic or mild form of the disease, a ruptured membrane, and cesarean. The laboratory results highlight the inconstant presence of some changes found in the list of potential predictors of the severity of the infection: Lymphopenia, high values of C-reactive protein, D-dimer, fibrinogen, platelets, Aspartate Aminotransferase, Lactate dehydrogenase, and ferritin. The study’s conclusion of this small group suggests that there may have been an intrauterine infection in late pregnancy and described characteristics of the pregnant women. Possible risk factors that could encourage vertical transmission have been identified.

## 1. Introduction

For almost two years, humankind has been facing a contagious infectious disease. The causative infectious agent is the Severe Acute Respiratory Syndrome Coronavirus 2 (SARS-CoV-2) [1]. Most cases diagnosed with Coronavirus Disease 2019 (COVID-19) had a favorable evolution, with mild to moderate respiratory symptoms, most of the time not requiring special treatment. However, cases with severe diseases requiring sustained medical intervention have been described, especially in populations with comorbidities [2].

Up to 30 January 2022, according to the weekly national surveillance report of COVID-19 infection published by the National Center for Surveillance and Control of Contagious Diseases from Bucharest, Romania, 2,216,525 people were diagnosed with COVID-19 in Romania [3]. Amongst them were pregnant women and even newborns; however, there is no particular national report concerning these two special populations.

SARS-CoV-2 infection transmission from mother to fetus may occur during pregnancy (congenital infection), at birth (intrapartum infection), or after birth (postpartum infection) [4,5,6]. Generally, vertical transmission of a virus could happen transplacental, during birth, or after birth through breastfeeding, depending on the moment of infection. The studies highlight that samples were taken from the nasopharyngeal level, breastmilk, vagina, and rectum of pregnant women, as well as amniotic fluid, to document the vertical transmission of the infection placenta and cord blood. In addition, oropharyngeal swabs, gastric juice, blood, feces, and urine samples were collected from neonates [7,8,9,10,11]. The clinical characteristics of pregnant women with SARS-CoV-2 infection and the vertical transmission potential of COVID-19 are unknown. Therefore, it is essential to study the potential risk of the infection on the fetus and newborn to implement adequate treatment [8,12,13]. However, in the third trimester, the studies described it as a rare phenomenon [7,8]. Other studies show no evidence for antepartum vertical transmission in women diagnosed with COVID-19 in late pregnancy [8,14].

Certain risk factors, such as cardiovascular diseases, diabetes, thrombophilia, coagulation disorders, and low immunity following pathological or physiological conditions (pregnancy), were associated with a worse prognosis if an infection is present [15]. However, the data are conflicting as far as a maternal or fetal infection is concerned [16,17].

There is much heterogeneity as far as vertical transmission is concerned. Complex molecular and immunological aspects can shed some light on this matter [18]. The angiotensin-converting enzyme (ACE) is reported to be abundant at the level of the syncytiotrophoblast, especially in the third trimester. Transmembrane protease serine 2 (TMPRSS2), furin, and cathepsin L are vital components that aid the virus’s internalization at the placenta level [7]. Furthermore, an imbalance of disintegrin and metalloproteinase (ADAM) 17 can affect the placenta’s capacity to shed the virus [19]. Fenizia investigated the possible genetic factors that modulate or do not modulate SARS-CoV-2 transplacental transmission. Due to an alteration in the inflammatory gene expression, the inflammatory cascade is more aggressive compared to uninfected profiles [20]. This observation agrees with Demirjian’s hypothesis that in utero transmission is more likely when high viremia is associated with extensive inflammation [4]. It is postulated that inflammatory and infectious conditions such as chorioamnionitis can favor vertical transmission of the virus [21].

In such cases, the evolution of pregnancy can have a poor outcome. Spontaneous abortions, stillbirths, and premature birth were associated with COVID-19 [22]. Information about the harmful effects of the infection on the general population, and even more so on pregnancy, to which pregnant women have access through the media, inevitably triggers feelings of anxiety and worries [23,24]. The paper describes clinical, laboratory, and imaging findings and potential vertical transmission of COVID-19 infection in three pregnant women. In addition, there were potential determinants of mother-to-child infection highlighted in cases.

## 2. Case Series Presentation

We identified three pregnant women diagnosed with COVID-19 whose newborns tested positive in the first 24 h after birth. They were hospitalized in the Obstetrics and Gynecology Departments of the County Emergency Clinical Hospital from Timisoara—one of the most significant tertiary units in the West Region of Romania. Between 1 April 2020 and 31 December 2021, the medical unit worked as COVID-support maternity. During this time, 281 pregnant women were delivered, and as a result, 283 newborns (2 twins).

At the moment of admission, nasopharyngeal (NP) and oropharyngeal (OP) swabs were taken from the pregnant women to diagnose SARS-CoV-2 infection. Immediately after birth, NP swabs were collected from all the newborns whose mothers were infected in the delivery room. The newborns had been placed in a transition ward until the test result was reached. After the test results, the positive newborns were placed in the ward for isolation. The parturient women were followed up in the infectious isolation ward. Clinical examination, vital function monitoring, and laboratory tests were performedfor mothers and neonates.

### 2.1. Specimen Collection and Virus Detection

From the pregnant women, OP and NP probes were collected. The specimens were combined at the collection site into a single transport vial. We also took a sample from the vagina and rectum of the pregnant women. Only NP probes from the newborns were collected during the first 24 h of life. The swabs were immediately placed into sterile tubes containing 2–3 mL of viral transport media and then were sent to the laboratory. The collected samples were sent to specialized units, named Technical Norms, which implemented national health programs. All personnel involved in collecting, transporting, and processing samples in the laboratory strictly observed the precautions necessary to prevent SARS-CoV-2 diseases, developed by the National Center for Surveillance and Control of Communicable Diseases from Bucharest, Romania [3]. According to the recommendations of the same Center, the samples were processed by methods to amplify nucleic acids and detect the virus—Reverse transcription polymerase chain reaction (RT-PCR). No samples were taken from the placenta, amniotic liquid, or umbilical cord. No COVID-19 antibodies were dosed in these cases on the pregnant women. Instead, anti-SARS-CoV-2 TOTAL (IgA, IgM, IgG) antibodies and specific immunoglobulin G (IgG anti SpikeS1) were determined in newborns.

### 2.2. Ethics Approval

Each pregnant woman was informed about all the aspects of the informed consent form regarding biological sampling, pregnancy monitoring, birth care, postpartum management, and SARS-CoV-2 infection management on the recommendation of the infectious disease specialist. As newborns’ legal representatives, mothers were informed and signed informed consent. Patients also agreed to participate in medical education and research. After being informed, the patients signed all the agreements. The study was conducted according to the guidelines of the Declaration of Helsinki and approved by the Ethics Committee of the County Emergency Clinical Hospital from Timisoara, Romania (protocol code 272/14 October 2021).

### 2.3. General and Clinical Characteristics of the Studied Pregnant Women

The pregnant women were diagnosed with COVID-19 in their third trimester. Two of them gave birth in our hospital (Pregnant 1 and 2), and one (Pregnant 3) was transferred after delivery from a secondary maternity hospital.

The clinical patients’ characteristics at admission are presented in Table 1. All pregnant women were in the third trimester of gestation—38 and 39 weeks. Their ages were between 21 and 35 years. Two pregnant women were from an urban environment and one from a rural environment. The patients were white Romanians with no previous pregnancy. In our studied group, intrauterine growth restriction, oligohydramnios, fetal anomalies, preterm premature rupture of membranes, preterm labor, preterm delivery, chorioamnionitis, or smoking history were not found. Only one patient had comorbidities: Pregnancy-induced hypertension (PIH) and thrombophilia. None of the patients were given the SARS-CoV-2 vaccine. Patient 1 was tested for SARS-CoV-2 because of the suggestive symptomatology—a cough. The other two patients were asymptomatic, so they had universal testing indications. The fetal presentations were cephalic in all cases. No evidence of prolonged labor was identified. Two of the patients (Pregnant 1 and 3) gave birth through a cesarean section with spinal anesthesia for the indications: Cervical dystocia (Pregnant 2) and fetal distress (Pregnant 3). Only the patient that gave birth vaginally presented moderate bleeding intrapartum. In the other cases, intrapartum bleeding was appreciated to be expected.

### 2.4. Laboratory Data, Chest X-ray, Treatment, and Outcome at Follow-Up

RT-PCR SARS-CoV-2 from OP and NP swabs, collected at admission, was positive for all patients. The vaginal swab was positive in the case of Pregnant 1, and the rectal swab was negative. Both were negative in the case of Pregnant 2. Pregnant 3 was delivered to another medical unit and came to our unit after birth; therefore, no swab was performed on the vagina and rectum.

After admission, peripheral blood was collected for laboratory analysis. Complete hemogram, coagulation, inflammation, liver, and kidney function tests were performed. Table 2 contains relevant laboratories at the moment of admission and on day 6 of hospitalization. Transitory leukocytosis was found in two patients. Transitory lymphopenia was documented by the values of the absolute lymphocyte count in two cases. C-reactive protein (CRP) had a high value in all patients. One patient had a mild form of anemia. Platelets (PLT) value was in the normal range in patients 1 and 2. However, it was elevated at Pregnant 3. Lactate dehydrogenase (LDH) was elevated in two cases. Ferritin (FER) levels had an increased value at Pregnant 1. D-dimers value was also elevated in two patients. Fibrinogen (Fg) was also found modified in two patients. As for hepatic function, only Aspartate Aminotransferase (AST) levels were elevated in one patient. Blood glucose (BG) levels were normal.

No patients had modified chest X-rays. Patients did not receive antiviral therapy or antibiotic therapy, or corticosteroids. Oral vitamins and micronutrients (Zinc 50 mg, C Vitamin 1000 mg/day, D Vitamin 4000 U/day) were administered to all patients. Both patients were delivered by C-section (Patients 2 and 3). According to the hospital protocol of thromboembolism prophylaxis in the postpartum period, they followed a treatment with low molecular weight heparin after birth. The patient with PIH was treated with methyldopa because of the high values of arterial blood pressure. The evolution of the cases was favorable in all cases. They were discharged on day 7. The patients were followed-up according to the monitoring procedure of those infected with SARS-CoV-2. After two weeks, mothers and newborns were in good condition.

### 2.5. Clinical Features of the Positively Tested Newborns Delivered by SARS-CoV-2 Infected Mothers

All newborns were full-term infants, classified based on the ponderal index appropriate for gestational age. Of all the newborns, only Newborn 1 showed symptoms. His mother also had an average form of illness. This newborn had generalized cyanosis at birth. Therefore, the Apgar score was 8 at 1 min. The newborn developed mild respiratory distress syndrome (RDS), tachypnea, minimal intercostal retraction, and audible expiratory grunt. Supplemental oxygen was applied for two days. Because RDS limited his enteral nutrition, parenteral nutrition was performed. Newborn 1 also received broad-spectrum antibiotic therapy for three days. The other two newborns did not present abnormal oxygen saturation and heart rate pathological changes. Fever was not present in these cases. Specific immunoglobulin G (IgG anti SpikeS1) had a positive level only in Newborn 3: 187 BAU (>17.8 BAU = POSITIVE). This value suggests in utero antibody transfer through the placenta—proof of intrauterine transmission. Anti-SARS-CoV-2 TOTAL (IgA, IgM, IgG) and specific immunoglobulin G (IgG anti SpikeS1) were unreactive in newborns 1 and 2. The evolution was favorable for all newborns (Table 3). They were discharged after seven days of hospitalization and continued isolation at home.

## 3. Discussion

Generally, vertical transmission of the virus could happen transplacental, during birth, or after birth through breastfeeding, depending on the moment of transmission [10]. There is evidence that infection with COVID-19 can occur in any way. In utero transmission is favored by the presence of angiotensin-converting enzyme receptors located at the level of the placenta [5]. In the intrapartum period, the fetus could be exposed to the virus located in vaginal fluids, as well as in maternal feces (during vaginal birth) [25] or the peritoneal fluids (during the cesarian section) [26]. Following WHO recommendations, reported evidence of the presence of the virus in the amniotic fluid or placenta is compelling for proving transplacental transmission [27]. Samples from the amniotic fluid and placenta would have been helpful in order to demonstrate the transplacental passage of the virus.

In this study, we cannot state with certainty that the viral infection occurred in utero or intrapartum. RT-PCR SARS-CoV-2 from OP and NP swabs was positive in all pregnant women. The first patient delivered vaginally, and the other two had C-sections. No evidence of prolonged labor was identified. Pregnant 3 had ruptured membranes for 3 h. The RT-PCR SARS-CoV-2 from the vaginal swab was positive in the case of Pregnant 1, and the rectal swab was negative. Both were negative in the case of Pregnant 2. No swab from the vagina and rectum was available from Pregnant 3. No samples from the amniotic liquid, placenta, cord blood, or breast milk were collected for financial reasons.

In newborns 1 and 2, Anti-SARS-CoV-2 TOTAL (IgA, IgM, IgG) and specific immunoglobulin G (IgG anti SpikeS1) had unreactive values. Both newborns had positive RT-PCR SARS-CoV-2 from NP swabs. This work complements our previous study, where the characteristics of newborns are described in more detail [28]. Since Patient 1 had a vaginal delivery, and we have no samples from the amniotic fluid or placenta to sustain intrapartum transmission, and we appreciate that it could be an intrapartum transmission. Although in the second case, the pregnancy was completed by cesarean section, with no samples from the amniotic fluid or placenta, we also classified this case as intrapartum transmission. Specific immunoglobulin G (IgG anti SpikeS1) had a positive level only in Newborn 3: 187 BAU (>17.8 BAU = POSITIVE). This value suggests possible in utero antibody transfer through the placenta. Therefore, we specify that Patient 3 was not vaccinated. In this case, it may have been an intrauterine or congenital infection.

Identifying the presence of IgM and IgG immunoglobulin for SARS-CoV-2 in newborn blood has been the subject of studies evaluating the possibility of vertical transmission. It is known that IgG, having a low molecular weight, passively crosses the fetoplacental barrier starting with the end of the second trimester of pregnancy. At birth, it can reach high values. Instead, IgM cannot cross it, being a large macromolecule [29]. More reports are related to the presence of the SARS-CoV-2 antibodies in newborns. In his study on 17 pregnant women who tested positive for COVID-19, Cosma et al. highlighted that their newborns developed IgG antibodies [30]. Another study that was suggestive of vertical or peripartum transmission of the infection described two newborns delivered by C-section with positive RT-PCR SARS-CoV-2 [31]. Another study presents a case report with possible vertical transmission. The newborn had positive SARS-CoV-2 IgM antibodies at 2 h of life, and his mother was suggestive of in utero infection [6].

The rate of COVID-19 vertical transmission has changed since the beginning of the pandemic. A review from 2021 declares a rate of 3.2% [32]. Kalamdani’s (2020) study declares that 7.74% of babies born from positive mothers were positive [33]. In another study from 2021, the reported rate of possible transmission was 11.9% [34]. If we refer to the total number of newborns born in the identified period, the presented three positive cases represent 1.06%.

The current literature is limited to the possible factors that encourage vertical transmission. However, the severity of the mothers’ symptoms does not seem to determine the occurrence of vertical transmission. In her prospective study, Jacob et al. found that out of the total number of positive pregnant patients that delivered positive babies, only 8.5% were symptomatic [26]. Kumar et al. report that most newborns with a positive COVID-19 test had asymptomatic mothers [35]. The duration of fetal exposure to SARS-CoV-2 can predict newborn infection. Furthermore, the shorter the interval between the mother’s first symptoms and delivery, the higher the risk of vertical transmission [36]. The studies described the general syndrome as fatigue, fever, respiratory syndrome, and gastrointestinal symptoms, which may be signs of severe disease progression [37]. The National Institutes of Health (NIH) grouped SARS-CoV-2 infection into the following severity of illness categories: Asymptomatic, mild illness, moderate illness, severe illness, and critical illness [38]. In our study, only Pregnant 1 had symptoms one day before hospitalization. According to NIH, she was categorized with a mild form of COVID-19 (patients with more COVID-19-specific signs and symptoms but do not have shortness of breath, dyspnea, or abnormal chest imaging). The other two patients were asymptomatic. None of the newborns from mothers diagnosed with severe COVID-19 tested positive for RT-PCR SARS-CoV-2 at birth. Studies revealed that a severe form of the mother’s disease did not increase the risk of vertical transmission [39,40]. In our case, although patients with a severe form of COVID-19 who gave birth were hospitalized during the studied period, vertical transmission was not present in any of these cases. The newborns were the same as the mother in the group studied with vertical transmission. The clinical profile of these patients showed that all were primigravida, nulliparous, and in the third trimester of pregnancy. In two cases advanced maternal age was found.

COVID-19 in pregnancy seems to be accompanied by a low level of certain micronutrients [41]. Therefore, the prophylactic and curative administration of vitamin C, vitamin D, and Zinc is a popular method of stimulating immunity in these cases [42]. Therefore, all patients in our study received treatment with supplements containing these microelements.

Different comorbidities can influence vertical transmission [5,39]. For example, thrombophilia is a condition of hypercoagulability [43]. Pregnancy increases this risk. Therefore, a pregnant woman with thrombophilia diagnosed with COVID-19 needs more attention to avoid thrombotic complications. Furthermore, the presence of PIH in infected patients seems to increase the risk for adverse pregnancy outcomes [44]. In the study group, Pregnant 3 was diagnosed with PIH and thrombophilia. Therefore, she received antihypertensive and anticoagulant treatment during pregnancy. We note possible factors that encourage vertical transmission: PIH, thrombophilia, asymptomatic, cough, an asymptomatic or mild form of the disease, a ruptured membrane, and cesarean. The study did not identify the presence of smoking history, vaccine before admission, atypical presentation, fever, or CXR abnormalities.

If we refer to the laboratory results, the studies describe the subsequent investigation to be used as a potential predictor of the severity of the infection: Albumin, AST, creatinine, D-dimer, fibrinogen, neutrophils, procalcitonin, and platelets [37]. Another interesting observation is that in severe forms of COVID-19, with oxygen dependence, a significantly higher AST was found but not ALT [45,46]. Pregnant 1 from our study group had AST more than twice elevated.

Most COVID-19 infections in pregnancy with vertical transmission are described in the third trimester [32]. In the current study, all three patients were 38–39 weeks. The RT-PCR method established the COVID-19 diagnosis. OP and NP samples were taken from the mothers and only NP from the newborn. Viremia was not performed in these cases.

Lymphopenia is usually described in SARS-CoV-2-infected individuals [47,48]. In addition, absolute lymphocyte count can be used as a marker of adverse clinical outcomes and disease severity in patients with COVID-19 [48]. In our study, lymphopenia was transitorily present in two cases. As far as inflammation is concerned, the paper investigated some inflammatory markers (D-Dimers, fibrinogen, and protein C reactive). The patient who delivered vaginally had an elevated CRP, and another had elevated D-Dimers.

Barrett et al. demonstrated that assessment of the platelets is essential in patients with COVID-19 [49]. They observe a trend toward less critical illness in positive patients with elevated platelets. In our study group, two patients presented with high values.

More studies have revealed that an elevated Lactate dehydrogenase (LDH) level is associated with a severe evolution of cases infected with SARS-CoV-2 [50,51]. However, although two pregnant women from our group had elevated values, their evolution was favorable without complications.

It seems that in the cases diagnosed with COVID-19, more patients present an increased ferritin level. This observation suggested using it as a predictor of disease severity [52,53]. Pregnant 1 from our study group presented an elevated serum ferritin value at admission. Hyper inflammation defined as CRP > 15 mg/dL or ferritin > 1500 µg/L [54] was not described. The hemoglobin level was lower in one case.

The laboratory results highlight the inconsistent presence of certain changes found in the list of potential predictors of the severity of the infection: Lymphopenia, high values of CRP, D-dimer, fibrinogen, platelets, AST, LDH, and ferritin.

Our findings highlighted that pregnant women with COVID-19 from the study, with possible vertical transmission, have a similar pattern of clinical characteristics to other pregnant or non-pregnant women in light of a medium form of the infection [55].

### Limitations

Limitations primarily involve the lack of generalizability because there were only cases. The implications in clinical practice are also limited because of the same reason. The newborns were tested in the first 24 h of life only by an NP swab. Vaginal and rectal swabs from the mothers were available in only two cases. In this study, we did not perform umbilical cord blood, amniotic fluid, and placental examination for the presence of the infection with SARS-CoV-2. Studies that include a larger population with more varied characteristics are required in the future.

## 4. Conclusions

SARS-CoV-2 infection occurred in the third trimester of pregnancy and had minor consequences for mothers and newborns. Vertical transmission is rare and can be established if there is enough information to prove it. Although our small sample limits our conclusions, we believe that our findings are essential for understanding the clinical, laboratory, and imaging characteristics of pregnant women diagnosed with COVID-19 and the potential for vertical transmission. It highlighted several possible factors that could encourage vertical transmission. A future study should look at the maternal profile of pregnant women with SARS-CoV-2 infection with proven vertical transmission compared to pregnant women with no vertical transmission.

## Figures and Tables

**Table 1 ijerph-19-10916-t001:** Patients’ characteristics and clinical details at admission.

Variable	Pregnant 1	Pregnant 2	Pregnant 3
**Demographics**			
Maternal age, year	35	21	35
Urban/Rural	U	R	U
Ethnicity	RO	RO	RO
Race	White	White	White
BMI (kg/m^2^)	26.72	26.03	27.68
**Antenatal and obstetric clinical data**			
Gravidity	1	1	1
Parity	0	0	0
Gestational age at delivery, wks	38	39	39
IUGR	N	N	N
Oligohydramnios	N	N	N
Fetal anomalies	N	N	N
PPROM	N	N	N
Preterm labor	N	N	N
Preterm delivery	N	N	N
Chorioamnionitis	N	N	N
Comorbidities	N	N	PIH, thrombophilia
Smoking history	N	N	N
**SARS-CoV-2 characteristics**			
**COVID-19 vaccine prior to admision**	N	N	N
Indication for testing	COVID-19 symptoms	universal testing	universal testing
** *Dispo* **	AP	AP	PP
Asymptomatic	N	Y	Y
Atypical presentation	N	N	N
Fever	N	N	N
Cough	Y	N	N
Form of disease	mild	asymptomatic	asymptomatic
** *Labor and Type of Birth* **			
Any anesthetic	N	spinal	spinal
Prolonged labor	N	N	N
Type of birth	vaginal	C-section	C-section
Indication of C-Section	-	Cervical dystocia	Fetal distress
Ruptured membrane	at the moment of delivery	N	>3 h
Cesarean recommendation	NA	cervical dystocia	fetal distress
Intrapartum bleeding	moderate	normal	normal

Abbreviations: Urban, U; Rural, R; RO—Romanian; BMI, body mass index (calculated as weight in kilograms divided by square of height in meters); wks, weeks; IUGR, intrauterine growth restriction; N, No; Y, Yes; PPROM—Preterm premature rupture of membranes; PIH, pregnancy-induced hypertension; SARS-CoV-2, severe acute respiratory syndrome coronavirus 2; COVID-19, Coronavirus Disease 2019; Dispo, initial disposition: AP, Admission (antepartum)/PP, Admission (postpartum); C-section, Caesarean section; NA, not applicable.

**Table 2 ijerph-19-10916-t002:** Available relevant laboratory data, treatment, and outcome at follow-up.

Variable	Pregnant 1	Pregnant 2	Pregnant 3
Relevant Laboratories at the Time of Admission/Day 6 of Hospitalization			
WBC (4–9.5 × 10^3^/μL)	7.8/7.14	10.36/NA	22.41/9.20
LYM (0.8–3.8 × 10^3^/μL)	0.85/1.49	1.00/NA	3.26/2.50
LYM (20–40%)	10.90/20.82	9.66/NA	14.55/27.30
CRP (0–10 mg/L)	69.8/14.60	14.60/NA	18/NA
Hgb (11.5–15 g/dL)	12.11/8.36	12.66/NA	10.90/10.30
PLT (150–400 × 10^3^/μL)	158.8/513.80	215.3/NA	470.20/412
LDH (120–246 U/L)	639/315	229/NA	369/NA
FER (4.5–170 μg/L)	284/34.60	10/NA	29.60/NA
D-dimer (0–243 ng/)mL	NA/377	1013/NA	NA/NA
Fg (200–393 mg/dL)	NA/278	587/NA	572/450
AST (14–36 U/L)	92/33	27/NA	34/29
BG (75–110 mg/dL)	68/73	87/NA	82/63
RT-PCR SARS-CoV-2	POSITIVE/NA	POSITIVE/NA	POSITIVE/NA
**CXR at admission**	Normal	Normal	Normal
**Specific treatment for COVID-19**	None	None	None
**Other treatment**	vitamins	vitamins, LMWH	vitamins, LMWH, methyldopa
**Outcome**	Discharged on day 7. Favorable evolution after two weeks of follow-up	Discharged on day 7. Favorable evolution after two weeks of follow-up	Discharged on day 7. Favorable evolution after two weeks follow-up

Abbreviations: WBC, White Blood Cell; LYM, Lymphocyte; CRP, C-reactive protein; Hgb, Hemoglobin; PLT, Platelets; LDH, Lactate dehydrogenase; FER, Ferritin; Fg, Fibrinogen; AST, Aspartate Aminotransferase; BG, Blood glucose; RT-PCR, Reverse transcription polymerase chain reaction; CXR, Chest X-ray; LMWH, low molecular weight heparin.

**Table 3 ijerph-19-10916-t003:** Clinical features of newborns.

Variable	Newborn 1	Newborn 2	Newborn 3
Weight at birth (g)	2990	3270	2900
Length at birth (cm)	50	52	50
Ponderal index (PI)	2.39	2.32	2.32
Classification based on PI	AGA	AGA	AGA
Gender	male	male	male
Apgar Score at 1 min	8	9	9
Apgar Score at 5 min	10	10	10
**Outcome**	Discharged to home in DOL 7	Discharged to home in DOL 7	Discharged to home in DOL 7

Abbreviations: g, gram; cm, centimeter; PI, Ponderal index; AGA, Appropriate for Gestational Age; DOL, Day of Life.

## Data Availability

The data presented in this case series are available on request from the corresponding author.

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
