# Peer review of "Clinical, Laboratory, and Imaging Findings of Pregnant Women with Possible Vertical Transmission of SARS-CoV-2—Case Series"

_ijerph, 2022, doi:10.3390/ijerph191710916_

Round 1

Reviewer 1 Report

It is a manuscript describing 3 cases of pregnancy with SARS-CoV2 newborn infection diagnosed at birth.  The study aimed to highlight the clinical, laboratory, and imaging findings of the pregnant women with confirmed COVID-19 with possible vertical transmission and identify possible factors that encourage vertical transmission.

This study lack consistency because of the very small number of cases without variables to analyze.

2 patients were symptomatic and one patient was asymptomatic. There are no specific modifications and particularities for clinical or laboratory findings.

One newborn, delivered at 38 weeks from the symptomatic patient developed mild respiratory distress syndrome, tachypnea, minimal intercostal retraction, and audible expiratory grunt, but his evolution was favorable. 

Author Response

Thank you for taking the time to review this article and for your valuable comments. Please find below the responses to the comments you sent us.

Extensive editing of English language and style required.

A native English-speaking colleague checked the manuscript.

Does the introduction provide sufficient background and include all relevant references? Can be improved

We complete the introduction and include more relevant references.

Are all the cited references relevant to the research? Must be improved

More relevant references were cited.

Is the research design appropriate? Must be improved

The design of the article was changed. The structure includes the following sections: 1.Introduction; 2. Case series presentation; 3. Discussion

Are the methods adequately described? Must be improved

The article's design has been modified into a more suitable form for a case series presentation.

Are the conclusions supported by the results? Must be improved

Since the article presents a series of three cases, it was not a significant statistical study. We have outlined the conclusions at the end of the Discussion section.

This study lack consistency because of the small number of cases without variables to analyze.

Because of the small number of cases, we present the results as a case series. We intend to soon collaborate with other medical units to increase the number of cases to realize an analysis of the variables.

2 patients were symptomatic, and one patient was asymptomatic. There are no specific modifications and particularities for clinical or laboratory findings.

The study aimed to highlight the clinical, laboratory, and imaging findings of the pregnant women with confirmed COVID-19 with possible vertical transmission and identify possible factors that encourage vertical transmission. Unfortunately, we can not have a significant conclusion because of the small number of cases. Nevertheless, we observed the characteristics of these cases.  

One newborn, delivered at 38 weeks from the symptomatic patient, developed mild respiratory distress syndrome, tachypnea, minimal intercostal retraction, and audible expiratory grunt, but his evolution was favorable. 

Under treatment, the evolution of the newborn with the mild form of the disease was favorable. We found it interesting that the patient with a mild form of the disease gave birth to a child who also developed a mild form of the disease. The children of the other two asymptomatic patients were asymptomatic. Therefore, we could suspect that the severity of the disease may be the same for newborns and mothers. Since the number of cases with positive newborns at birth was small and since we had patients with a severe form of the disease and their children were negative at birth, the fact supports that the hypothesis is false.

Thank you for your cooperation and support.

Reviewer 2 Report

Dear Author

-You didn't cited the paper Diagnostics 2022, 12, 1668. https://doi.org/10.3390/diagnostics12071668 about the three neonate with vertical transmission.

-Also the reference 5 is not cited at line 57

-You have not demonstrated that the transmission was vertical whereas in your previous article you explained the vertical transmission for your newborns

-risk factors such as thrombophilia are considered factors of severity of attack by covid with increased risk of coagulation, but you have not demonstrated how they can be considered risk factors for vertical transmission of COVID

Author Response

Thank you for taking the time to review this article and for your valuable comments. Please find below the responses to the comments you sent us.

Are all the cited references relevant to the research? Must be improved

More relevant references were cited.

-You didn't cited the paper Diagnostics 2022, 12, 1668. https://doi.org/10.3390/diagnostics12071668 about the three neonate with vertical transmission.

The article describing the neonates' characteristics was published on 9 July, 2022. I uploaded this paper on 3 July, 2022. I cited it in this revised form of the paper.

-Also reference 5 is not cited in line 57

In line 57, the references are cited from 4 to 6. We used IEEE citation style, which notes in this case [4] – [6].

-You have not demonstrated that the transmission was vertical, whereas in your previous article, you explained the vertical transmission for your newborns.

The previous article is the point of view of neonatologists. This article is the point of view of obstetricians. We collaborated on both articles, but we prefer to respect the point of view of each specialist, especially since some aspects related to the COVID infection are not clarified.

-risk factors such as thrombophilia are considered factors of severity of attack by covid with increased risk of coagulation, but you have not demonstrated how they can be considered risk factors for vertical transmission of COVID

We observe that one of the patients is diagnosed with thrombophilia, and we note that it can be a risk factor. However, with one single case, we cannot prove anything. It is just a supposition.

Thank you for your cooperation and support.

Reviewer 3 Report

Thanks, Authors,  for their submission: Clinical, Laboratory, and Imaging Findings of the Pregnant 2 Women with Possible Vertical Transmission of SARS-CoV2 - 3 Case Series. 

The study aimed to highlight the clinical, laboratory, and imaging findings of the pregnant women with confirmed COVID-19 with possible vertical transmission and identify possible factors that encourage vertical transmission. Between April 01, 2020, and 22 December 31, 2021, 281 pregnant women diagnosed with Coronavirus Disease 2019 (COVID-19) gave birth in the Obstetrics and Gynecology Departments of the tertiary unit of County Emergency 24 Clinical Hospital from Timisoara. Three newborns (1.06%) tested positive. The characteristic of 25 these three cases was described as a short series. In two cases, the patients were asymptomatic. In one case, the patient developed a mild form of COVID-19 with a favorable evolution in all cases. 

- Generally, vertical transmission of the virus could happen transplacental, during birth, or after birth through breastfeeding, depending on the moment of transmission. On this point, I would suggest considering this recent paper:  From Undetectable Equals Untransmittable (U=U) to Breastfeeding: Is the Jump Short? Prestileo, T., Adriana, S., Lorenza, D.M., Argo, A. Infectious Disease Reports, 2022, 14(2), pp. 220–227

-at line: "In the intrapartum period, the baby could be exposed to the virus: I suggest changing the word "baby" and substituting it with "fetuses". 

- as the Authors affirmed (lines 219-221):  "Samples from the amniotic fluid and placenta would have been helpful in order to demonstrate the transplacental passage of the virus.  In this study, we cannot state with certainty that the viral infection occurred in utero or intrapartum". I would like to know why this test was not carried out. 

 - According to the hospital protocol of thromboembolism prophylaxis in the postpartum period, they followed a treatment with low molecular weight heparin after birth. The patient with PIH has been treated with methyldopa also:  please specify the rationale of therapy. 

- "His mother also had an average form of illness":  the Authors should specify the meaning of average form; is this a classification by marks or clinical findings? 

Author Response

Thank you for taking the time to review this article and for your valuable comments. Please find below the responses to the comments you sent us.

English language and style are fine/minor spell check required.

A native English-speaking colleague checked the manuscript.

Are the results clearly presented? Must be improved

We made some changes in the results section. The article's design has been modified into a more suitable form for a case series presentation. The structure includes the following sections: 1.Introduction; 2. Case series presentation; 3. Discussion.

Are the conclusions supported by the results? Can be improved

Since the article presents a series of three cases, it was not a significant statistical study. We have outlined the conclusions at the end of the Discussion section.

- Generally, vertical transmission of the virus could happen transplacental, during birth, or after birth through breastfeeding, depending on the moment of transmission. On this point, I would suggest considering this recent paper:  From Undetectable Equals Untransmittable (U=U) to Breastfeeding: Is the Jump Short? Prestileo, T., Adriana, S., Lorenza, D.M., Argo, A. Infectious Disease Reports, 2022, 14(2), pp. 220–227

Thank you for the suggestion. I included this reference in the article.

-at line: "In the intrapartum period, the baby could be exposed to the virus: I suggest changing the word "baby" and substituting it with "fetuses". 

We changed it.

- as the Authors affirmed (lines 219-221):  "Samples from the amniotic fluid and placenta would have been helpful in order to demonstrate the transplacental passage of the virus. Unfortunately, in this study, we cannot state with certainty that the viral infection occurred in utero or intrapartum". I would like to know why this test was not carried out. 

At that time, the local protocol did not include these kinds of sampling, and there was no financial support.

 - According to the hospital protocol of thromboembolism prophylaxis in the postpartum period, they followed a treatment with low molecular weight heparin after birth. The patient with PIH has been treated with methyldopa also:  please specify the rationale of therapy. 

The patient diagnosed with PIH has a cardiological consult and the recommendation of methyldopa because of the high blood pressure values. I completed this reason in the article also.

- "His mother also had an average form of illness":  the Authors should specify the meaning of average form; is this a classification by marks or clinical findings? 

The classification of the severity of the illness follows the recommendations of the National Institutes of Health. Therefore, we complete this observation in the article.

Thank you for your cooperation and support.

Round 2

Reviewer 3 Report

Authors took into account suggestions, gave explanations for some laboratory details   and improved  their  research.  
